# RGB-Marking to Identify Patterns of Selection and Neutral Evolution in Human Osteosarcoma Models

**DOI:** 10.3390/cancers13092003

**Published:** 2021-04-21

**Authors:** Stefano Gambera, Ana Patiño-Garcia, Arantzazu Alfranca, Javier Garcia-Castro

**Affiliations:** 1Cellular Biotechnology Unit, Instituto de Salud Carlos III, 28220 Madrid, Spain; stefano.gambera87@gmail.com (S.G.); mariaaranzazu.alfranca@salud.madrid.org (A.A.); 2Department of Pediatrics, Laboratory of Advanced Therapies for Pediatric Solid Tumors, Solid Tumor Area, CIMA and Instituto de Investigación Sanitaria de Navarra, University Clinic of Navarra, IdiSNA, 31008 Pamplona, Spain; apatigar@unav.es; 3Immunology Department, Hospital Universitario de La Princesa, 28006 Madrid, Spain

**Keywords:** osteosarcoma, clonal evolution, RGB-marking, cancer models

## Abstract

**Simple Summary:**

Nowadays, we are assisting a re-discovered interest in the field of cancer heterogeneity and in defining the clonal dynamics governing tumor growth, progression, and therapy resistance. Sequencing data suggest different models of cancers development and a relationship with patients’ prognosis and therapeutic response. Only a few studies have attempted to reconstruct osteosarcoma evolution, providing evidence of linear and branched pattern of clonal development. In this study, we employed a single-cell marking strategy to study the clonal dynamics of human, canine, and murine osteosarcoma models. With our collection of primary samples and cell lines, we demonstrate that different clones can evolve in parallel and generate sub-clones with similar tumorigenic potential, in a sort of extremely branched development of neutrally coexisting clones.

**Abstract:**

Osteosarcoma (OS) is a highly aggressive tumor characterized by malignant cells producing pathologic bone; the disease presents a natural tendency to metastasize. Genetic studies indicate that the OS genome is extremely complex, presenting signs of macro-evolution, and linear and branched patterns of clonal development. However, those studies were based on the phylogenetic reconstruction of next-generation sequencing (NGS) data, which present important limitations. Thus, testing clonal evolution in experimental models could be useful for validating this hypothesis. In the present study, lentiviral LeGO-vectors were employed to generate colorimetric red, green, blue (RGB)-marking in murine, canine, and human OS. With this strategy, we studied tumor heterogeneity and the clonal dynamics occurring in vivo in immunodeficient NOD.Cg-Prkdcscid-Il2rgtm1Wjl/SzJ (NSG) mice. Based on colorimetric label, tumor clonal composition was analyzed by confocal microscopy, flow cytometry, and different types of supervised and unsupervised clonal analyses. With this approach, we observed a consistent reduction in the clonal composition of RGB-marked tumors and identified evident clonal selection at the first passage in immunodeficient mice. Furthermore, we also demonstrated that OS could follow a neutral model of growth, where the disease is defined by the coexistence of different tumor sub-clones. Our study demonstrates the importance of rigorous testing of the selective forces in commonly used experimental models.

## 1. Introduction

Cancer is considered an ecological and evolving disease driven by the reiterative accumulation of new genetic diversity, and selection of the most aggressive cancer variants [1,2]. With the introduction of next-generation sequencing (NGS) and single-cell sequencing, branched patterns of sub-clonal evolution and neutral dynamics of coexistence have also been proposed [3,4,5]. The linear, branching, and neutral models of cancer evolution differ in terms of the relationship of coexistence among cancer clones, which also define the therapeutic target [6,7]. Assessment of tumor evolutionary history through direct tumor sequencing represents only part of the ongoing clinical investigation [8,9]. Patient-derived material is also employed for studying tumor cells in vitro and in vivo [10]. A standard method to rigorously test sub-clonal evolution and selective events in laboratory models is still not rigorously employed, even if it could have tremendous implications in science reproducibility [11,12,13,14].

OS is the most common malignant bone tumor affecting children and adolescents [15] and has one of the most complex genomes among pediatric cancers [16,17]. Only a few studies have attempted reconstructing of OS evolution, providing evidence of linear and branched patterns of clonal development [18,19,20], genomic macroevolutionary events [21], and minor clones evolving therapy resistance [22]. However, those studies were based on the phylogenetic reconstruction of NGS data, which do present important limitations [4].

Employing genetically modified cells, we previously demonstrated that murine OS can follow a neutral model of growth [23]. In this study, we employed a colorimetric single-cell RGB-marking technique to study the clonal dynamics of murine, canine, and human OS of spontaneous etiology. Here, we demonstrate that tumor-propagating potential (TPP) is scarce in tumor cell lines, as well as in primary samples. Nevertheless, clones actively growing in vivo and so retaining TPP generate clonal populations composed of sub-clones with similar tumorigenic ability. This is in contrast to a linear and competitive model of evolution; our data demonstrate that OS can follow a neutral model where sub-clones do not outcompete each other.

## 2. Material and Methods

### 2.1. Osteosarcoma Cell Lines, Primary Samples Isolation, and Culture

K5 OS-50 cells were kindly provided by Dr. Silvia Miretti; the cell line was obtained by end-point dilution of a spontaneous murine OS occurring in the distal femur of a female BALB/c mouse, thus representing a monoclonal cell line [24]. Saos2 cell line was purchased from ATCC and represents a human OS cell line established from a spontaneous OS developing in a 21-year-old patient affected by a multifocal and metastatic osteoblastic OS; 531B corresponds to the chemo-naïve true-cut biopsy of the right distal femur lesion [25,26]. Laikos cells were isolated at Veterinary Hospital at Alfonso X University (Madrid, Spain), from a spontaneous canine chondroblastic OS case occurring in a female 8-year-old Belgium Shepherd after chemotherapeutic treatment (PALL/DOX/CFX). Laikos cells were isolated from fresh tumor tissue by mechanical and enzymatic dissociation with trypsin (0.25%)/EDTA (Lonza, Besel-Switzerland); enzymatic digestion was performed at 37 °C for 30 min. K5, Saos2, and Laikos cells were maintained in Dulbecco’s modified Eagle’s medium supplemented with 10% fetal bovine serum (FBS), 1% Pen/Strep, and 1% Glutamax (Lonza). The 531B cells were cultured in α-MEM with 10% FBS, pen/strep and 2 mM L-glutamine. Cell cultures were tested for mycoplasma contamination (MycoAlert-Mycoplasma Detection kit, LONZA). Human cell lines were authenticated by STR profiling.

### 2.2. LeGO-RGB Technology and Vectors Production

Multicolor tumor cell RGB-marking was achieved by employing LeGO-RGB lentiviral vectors; those vectors transduce cells at a highly different but constant level of red (mCherry), green (Venus), and blue (Cerulean) fluorescent proteins, generating an inheritable color range used as a color-guided clonal cell tracer [27,28,29]. LeGO-C2 (Red FP, Addgene: 27339), LeGO-V2 (Green FP, Addgene: 27340) and LeGO-Cer2 (Blue FP, Addgene: 27388) plasmids were kindly provided by Dr. Kristoffer Riecken. LeGO-RGB lentiviral vector packaging was obtained by transient calcium phosphate transfection in HEK-293T cell; those vectors are third-generation self-inactivating lentiviral vectors. Supernatant was collected 48 h after transfection and concentrated by ultracentrifugation.

### 2.3. Multicolor RGB Marking

Lentiviral transduction was performed on bulk cell lines and no endogenous surface markers were employed to pre-select tumor cells. Lentiviral particle mixtures were added to 1.5 × 10^5^ cells of each OS cell culture and incubated overnight. Efficient color marking estimation was assayed 3 days after transduction; for MOI estimation, see the specific results section in the text.

### 2.4. Flow Cytometers

RGB-fluorescence signal was analyzed using two different cytometers. In a BD FACSAria II cell analyzer (BD Bioscience, San Jose, CA, USA), Cerulean fluorescent protein was excited at 405 nm and detected with a 525/50 bandpass filter, Venus was excited at 488 nm and detected with a 530/30 bandpass filter, and Cherry was excited at 488 nm and detected with a 585/15 bandpass filter. In a BD LSRFortessa cell analyzer, Cerulean fluorescent protein was excited at 405 nm and detected with a 450/50 bandpass filter, Venus was excited at 488 nm and detected with a 530/30 bandpass filter, and Cherry was excited at 561 nm and detected with a 610/20 bandpass filter.

### 2.5. In Vivo Assays

Immunodeficient NOD.Cg-Prkdcscid-Il2rgtm1Wjl/SzJ (NSG) mice were employed as animal models to study murine, canine, and human OS development in vivo. RGB-marked cells were trypsinized, washed, and resuspended in PBS; 1.5 × 10^5^ cells were inoculated subcutaneously in the flank of anesthetized (2% isoflurane) 8–10-week-old mice; litters were randomized. A sample size of 5 mice for each experimental group was established, given that the study aimed to evaluate the dichotomy of monoclonality vs. polyclonality; sample size reduction/increase was dependent on cell line tumorigenicity. Alternatively, RGB-marked tumor cells were implanted orthotopically. Orthotopic procedure was performed by bending the mouse leg at 90° to drill the tip of the tibia with a 25 G needle. Following that, cell suspension was deposited in the medullar space with a 27 G needle. After tumor cells inoculation/implantation, mice were revised daily for their health status and tumor appearance; mice were sacrificed when tumor volume reached 1 cm^3^, researchers were not blind.

### 2.6. Tumor Clonal Composition Analyses

Explanted RGB-marked tumors were dissociated by mechanical and enzymatic dissociation with trypsin (0.25%)/EDTA (Lonza, Basel, Switzerland); enzymatic digestion was performed at 37 °C for 30 min. After cell attachment and a short in vitro culturing, cells were trypsinized and resuspended in PBS, then analyzed by flow cytometry. FCS data were loaded in FlowJO (FlowJO, Ashland, OR, USA) to study RGB-marker expression by standard 2D and 3D flow cytometry, and in Cytobank (Cytobank, Santa Clara, CA, USA), to study the clonal architecture of the samples performing different types of unsupervised analyses with high dimensional methods(https://www.cytobank.org/, access on 15 March 2018). Explanted RGB-marked tumor tissue was also fixed overnight in 4% formalin or 1% PFA, and then decalcified with Osteosoft (Merck-Millipore, Burlington, MA, USA) for 72 h before inclusion in optimal cutting temperature Tissue-Tek (Sakura Finetek, AJ Alphen aan den Rijn, The Netherlands). Those samples were processed for histologic staining and confocal fluorescence analysis by cryosectioning. Eight-micrometer slides were defrosted and stained according to histologic standards or pre-warmed, hydrated in PBS for 2 min, and then mounted using ProLong for confocal microscopy studies.

### 2.7. Unsupervised Clonal Tumor Architecture Analyses

Variants’ clustering of samples was performed taking into account the intensity of Cerulean, Venus, and Cherry signals. A visual stochastic network embedding (viSNE) map using t-distributed stochastic neighbor embedding (t-SNE) algorithms was generated [28]. Maps are provided as 2D scatter plots and indicate the 3 fluorescent channels’ (RGB) intensity. A spanning-tree progression analysis of density-normalized events (SPADE) algorithm was applied to reconstruct population hierarchies and visualize clonal populations in a tree-like structure [29]. SPADE performs agglomerative clustering, taking into account the intensity of selected channels in a density-dependent fashion, and down-samples to equally represent rare and abundant populations. In this case, SPADE was set to represent the data as 200 different clones.

### 2.8. Clones Isolation and Retracing

Three different clones, or populations of cells sharing the same combination and similar intensities of RGB colorimetric fluorescent markers, were isolated from RGB-marked primary tumors by flow automated cell sorting (FACS). FACS was performed using a SONY iCyt SY3200 Cell Sorter (SONY Biotechnology, San Jose, CA, USA). Cerulean fluorescent protein was excited at 405 nm and detected with a 525/50 bandpass filter, Venus was excited at 488 nm and detected with a 525/50 bandpass filter, and Cherry was excited at 532 nm and detected with a 615/30 bandpass filter. Sorting purity was verified by flow cytometry and confocal microscopy after a short in vitro expansion. LeGO-RGB vectors contain loxP sites flanking the provirus upon integration in host cell DNA, allowing fluorescent marker deletion by Cre recombinase. FACS-categorized clones were decolored in vitro by the adenoviral transduction of Cre recombinase. Decolored clones underwent a second round of lentiviral RGB-marking to clonally track each sub-clone; this process is simplified as recoloring in the text.

### 2.9. Confocal Microscopy Analyses

A Leica TCS-SP5 (Leica Microsystems, Wetzlar, Germany) microscope was employed in confocal studies. Images were obtained by maximum projection of a 10-layer stack of 8 µm thick sections; image processing was performed using Leica LAS AF (Leica Microsystems). In vitro confocal microscopic studies were performed by seeding cells in multi-chambers (EMD Millipore, Burlington, MA, USA) and incubating cells overnight. Then, slides were washed with PBS and fixed with 4% formalin or 1% paraformaldehyde (PFA) for 1 min, washed again with PBS, and mounted with ProLong (Life-technologies, Carlsbad, CA, USA).

### 2.10. Statistical Analyses

Data were graphed with GraphPad Prism (GraphPad Software, La Jolla, CA, USA). All the experiments were performed at least in triplicates. Correlation between population frequencies and days of culture was estimated by the Pearson coefficient of correlation with a confidence interval of 95%. *p*-values less than 0.05 were considered statistically significant.

## 3. Results

### 3.1. Murine OS Cell Line and Oligoclonal Tumors In Vivo

The first model of study was a monoclonal murine cell line (K5 OSA) obtained by single-cell plating of a spontaneous OS. K5 OSA shows a great morphological heterogeneity during in vitro culture, with the presence of fibroblastic shaped cells, tile-like osteoblastic cells, and star-shaped cells (Figure 1A). RGB lentiviral vectors were employed to mark K5 OSA and follow tumor cell development in vitro and in vivo. Transduction with RGB lentiviral vectors generates a highly variable and inheritable color range in target cells. Any targeted cell and its derived progeny constitute a population that will be defined as a clone in the text, because they share the constitutive expression of a common combination of colorimetric fluorescent markers. Thus, we first optimized the transduction protocol to obtain an appropriate full-spectrum of colors combination. RGB-marking was validated by flow cytometry and confocal microscopy following the principle of avoiding color saturation, to obtain a range of single color-marked cells (different tones of red, green, or blue), and their possible combinations. In this regard, K5 OSA initially showed a transduction efficiency of 55% at MOI 1 (Appendix A), but the best color spectrum was achieved by further reducing the number of viral particles to a MOI of 0.75 per vector (Figure 1B and Appendix A). Around 30–35% of cells were positive for each fluorescent marker or their possible combinations (Figure 1C and Appendix A). Those cells are named K5-RGB cells throughout the text.

Fluorescent marker expression and clonal composition were analyzed by flow cytometry during 35 days of K5-RGB in vitro culturing (Figure 1D). During the assay, 2D and 3D flow cytometry plots maintained a cloudy distribution of fluorescent marker colorimetric combinations, indicating no enrichment of any color-coded clones at the cost of the others (Appendix A). We also employed visual interactive Stochastic Neighbor Embedding (viSNE) and Spanning Tree Progression of Density Normalized Events (SPADE) analyses to simultaneously study our high-dimensional single-cell data. With both methods, we did not detect significant changes in the sub-clonal frequency of K5-RGB cells in vitro (Appendix A). K5-RGB cells were also inoculated in immunodeficient mice showing a 100% of tumor incidence (18/18) and with an average latency of 49 days (28–63 days). Independently of their ectopic (*n* = 15) or orthotopic (*n* = 3) location, the in vivo clonal composition of K5-RGB tumors was characterized by evident clonal enrichments. Confocal microscopy study revealed the development of different clones, each one growing in geographically separated patches except for sub-clonal mixing in the border zone of neighboring patches (Figure 1E and Appendix A). Tumors were also dissociated and cultured ex vivo. Two-dimensional and three-dimensional flow cytometry analysis of ex vivo cultured tumor cells confirmed the enrichment of some clonal populations, now becoming more evident due to the loss of clonal populations with a similar colorimetric markers’ combination (Figure 1F and Appendix A). Similar results were obtained by viSNE and SPADE analyses (Figure 1G,H), showing evident changes in tumor clonal composition (Appendix A).

In summary, we efficiently marked K5 OSA cells with LeGO-RGB lentiviral vectors and studied their growth in vitro and in vivo. While the polyclonal composition of K5-RGB cells was stable in vitro, only some clones could expand and generate tumors in vivo.

### 3.2. Human OS Cell Line and Oligoclonal Tumors In Vivo

To further test the relevance of our findings, we extended our clonal studies to human samples. Saos2 is a well-established OS cell line obtained from an 11-year-old female patient after treatment; this cell line presents a homogenous epithelial morphology when cultured in vitro (Figure 2A), and some reports indicate it is tumorigenic in vivo [30,31], depending on the mouse strain. Lentiviral LeGO-RGB vectors were employed to also transduce Saos2 cells and study their clonal dynamics in vitro and in vivo. RGB marking was achieved and validated by confocal microscopy and flow cytometry (Figure 2B,C). Efficient RGB marking was obtained firstly by calculating the specific cell line transduction efficiency at increasing MOI concentrations (Appendix A). Finally, the best fluorescent marking and color variability was achieved at a final MOI of two for the Cerulean vector, while keeping Venus and mCherry vectors to the same MOIs (0.75); those cells are further named Saos2-RGB in the text.

Clonal composition and fluorescent marker stability of Saos2-RGB cells were studied in vitro. Fluorescent marker expression was quantified by flow cytometry. Two-dimensional and three-dimensional flow cytometry plots maintained a heterogeneous distribution of fluorescent markers’ colorimetric combinations, indicating no differential enrichment of any color-coded clone in the context of the polyclonal cell line (Figure 2D and Appendix A). Moreover, viSNE and SPADE single-cell high-dimensional analyses confirmed the result (Appendix A). After in vivo inoculation, Saos2-RGB cells showed an incidence of 40% (2/5 mice) and a latency of 106 days in immunodeficient mice. Saos2-RGB tumor clonal composition was characterized by the enrichment of some clones. Indeed, confocal microscopy study indicated the evident development of some monoclonal areas with scant clonal mixing (Figure 2E). Clonal enrichment was further confirmed by 3D flow cytometry, where the distribution of fluorescent markers’ colorimetric combinations became less heterogeneous (Figure 2F). Thus, while in vitro cultures do not determine the enrichment of clones at the cost of the others, flow cytometry plots of Saos2-RGB explanted tumor cells were characterized by the emergence of some clones (Appendix A). This clonal dynamic was also confirmed in the tSNE and spanning tree graph, where entire areas or branches in the graphs were completely lost (Figure 2G,H). Additionally, viSNE and SPADE analyses identified the enrichment of those clonal populations effectively retaining TPP (Appendix A).

We repeated the same experiments of RGB-marking and in vivo testing using primary 531B cells, obtained from a primary tumor of a 21-year-old female multifocal OS patient, and primary Laikos cells, obtained from a primary tumor of an 8-year-old dog with OS. In contrast to Saos2 and K5 OSA cells, those primary cells are at a low passage of in vitro culturing after tumor explant. We marked these cells with RGB lentiviral vectors and inoculated them into immunodeficient mice. The 531B cells did not induce tumor formation at subcutaneous location, while exhibiting a low tumor incidence intramedullary. By contrast, Laikos-RGB cells were 100% tumorigenic (4/4), presenting a shorter latency of 46 days (40–52 days). However, also in these cases, confocal microscopy study indicated an oligoclonal tumor composition where both tumor types were composed of few clones (Figure 3).

### 3.3. Sub-Clonal Dynamics of Tumorigenic Clones and Neutral Growth Model

The obtained results indicated a difference between the polyclonal composition of cultures stably growing in vitro and the few clones effectively able to develop in vivo. Thus, we decided to test the sub-clonal dynamics occurring within a clone retaining TPP (Figure 4A). After 35 days of in vivo tumor development, K5-RGB tumor-derived clones were grouped by FACS depending on their RGB fluorescent markers’ expression (Figure 4B). FACS-separated tumor-derived clones were shortly expanded in vitro to check their clonal purity and establish a monoclonal cell line (Figure 4C). Interestingly, monoclonal cell lines were reproduced in vitro by their heterogeneous pre-implantation morphology. Three clones were transiently transduced with adenoviral vectors expressing Cre recombinase to target the LoxP sites carried by LeGO-RGB lentiviral vectors and eliminate their clone-specific RGB fluorescent markers (Figure 4D). We did not observe any toxic side effects, but one monoclonal population showed resistance to recombination and was excluded from the study. Recombined FACS-grouped clones were transduced with a second round of LeGO-RGB vectors (Figure 4E,F) to re-color the clonal population. Following that, re-colored clones were inoculated in vivo.

In this secondary transplantation, K5-RGB tumor clones developed faster, presenting a reduction in tumor latency to 20 days, and an incidence of 100% (6/6). Confocal microscopy study revealed a new dynamic of growth and, while some sub-clones dominated large areas, sub-clonal mixing was also evident (Figure 4G). Flow cytometry analysis presented a notably lower reduction in tumor clones (Figure 4H) and similar results were obtained by ViSNE and SPADE analyses (Figure 4I,J and Appendix A). Again, during in vitro culturing, no evident changes in their sub-clonal composition were observed (Appendix A).

Globally, these data are in frame with a model of neutral growth dynamic within cancer sub-clones at secondary transplantation.

## 4. Discussion

During recent years, cancer research has rediscovered interest in defining the models governing tumor growth and evolution [4,8,9]. Nowadays, cancer evolutionary patterns are considered to be of primary importance, because the therapeutic response mostly depends on this evolutionary process and could represent an exploitable therapeutic target [2,32,33]. Furthermore, different models of cancer growth are also related to patients’ prognosis, disease progression, and therapy response [4,9,34,35].

Animal models represent a largely exploited tool for studying human cancers. Nevertheless, xenografting could represent a strong experimental selective event for cancer cells. The empirical proof for this concept was shown with cancer stem cell frequency estimation, which can be higher under improved xenotransplantation conditions or in less immunodeficient mice [36]. Furthermore, genetic studies have suggested that patient’s derived xenografts present different genetic marks from those detected in tumors evolving in patients [13,37]. These results suggest that TPP could be reduced in a different biological system and point out the importance of the microenvironment for tumor cell development. Nevertheless, in vitro culturing and in vivo xenografting are not currently rigorously assayed in terms of clonality. Species-specific microenvironmental changes, tumor architecture destruction, extensive culturing with changes in nutrients, or monolayer growth could potentially alter the composition of cancer populations selecting some specific cancer variants.

In this study, we employed lentiviral-LeGO vectors to RGB-mark OS cells and study tumor heterogeneity and in vivo clonal dynamics. This technology is a powerful tool for clonal cell studies because it represents an unbiased approach for studying tumor physiology [23,38]. Here, we applied this method to study murine, canine, and human OS samples in vivo, including cell lines and primary cultures. Those samples represent cells at the final point of a tumorigenic process, and more specifically, we focused on cells with TPP and their clonal dynamic of growth. An important difference from previous studies is that with lentiviral LeGO- RGB marking, we can employ fluorescent protein colorimetric combinations to simultaneously study different tumor cells, and thus avoid biasing our study through previously published cancer stem cells markers [39]. Moreover, the most important limitations of this technology are related to the time needed to achieve a balanced transduction with lentiviral LeGO-RGB vectors, the use of confocal microscopy techniques for RGB marker heterogeneity validation, and the employment of immunodeficient mice to support the engraftment of non-murine cells. NSG mice lack mature T cells, B cells, NK cells, and present deficiencies of several cytokines’ pathways and defects of innate immunity, which can be extremely relevant in tumor physiology (see below).

In accordance with previous studies, we found that TPP was low among cultured tumor cells, with evidence of strong selective clonal events occurring at tumor transplantation. This conclusion is strengthened by the occurrence of clonal selection among different OS samples (murine, canine, and human OS), indicating the consistency of our findings. By contrast, in vitro culture conditions did not affect OS cells’ sub-clonal composition. However, this observation does not exclude an ongoing evolutionary process, which could even be detrimental in terms of TPP given a reduction in selective forces. Additionally, our results support that TPP is not limited to the clonal growth of a single dominant clone, and different tumor clones can drive tumor formation. This observation would exclude a linear model of tumor evolution, in favor of a branched model of tumor growth. Our conclusions are in agreement with previous studies where RGB-marking was applied to transformed human hepatocytes [40] and human neuroendocrine carcinoma cell lines [41]. Those studies show that tumors are composed of few clones, indicating a scarce TPP. Similarly, a reduction in genetic barcodes in tumors induced by head and neck squamous cell carcinoma cell line has been observed [42], and in agreement with our hypothesis, clonal heterogeneity is increased by the co-injection of cancer-associated fibroblasts. By contrast, results obtained with human primary colon carcinomas [43] and head and neck squamous cell carcinoma cell lines [42] indicate a more consistent clonal heterogeneity at primary transplantation. However, this is also in agreement with our hypothesis, because tumor cells were isolated as spheroid cultures or injected in gelatin matrix. Likewise, we recently demonstrated that clonally heterogeneous OS can be induced by injecting transformed murine mesenchymal stem cells into a bioengineered osteoinductive microenvironment [23].

Consequently, primary transplantation only represents the first step for studying the model governing OS growth, allowing us to isolate clones still retaining TPP. Indeed, clonal populations isolated from RGB tumors are composed of sub-clones with a similar tumorigenic potential at serial transplantation. This intra-clonal dynamic recapitulates our previous findings, showing that once an invasive OS clone develops, this is composed of a heterogeneous mix of sub-clones with similar TPP [23]. Taken together, our data indicate that OS can follow a pattern of neutral growth in vivo and that the selective events occurring at primary transplantation would be the outcome of microenvironmental and selective forces changes, which could even be detrimental in terms of TPP. Our model agrees with recent findings confirming a tendency to the sub-clonal diversification of OS cells, due to the existence of a complex ecosystem of tumor-associated cells actively participating in the development and progression of the disease [44]. Thus, the presented results and previously published studies are consistent with the crucial role of the microenvironment in the maintenance of TPP, the difficulty to rigorously mimic this original and highly heterogeneous ecosystem in experimental models, and the existence of multiple clones in the same tumor.

## 5. Conclusions

In summary, our study confirms the value of the lentiviral LeGO-RGB system in tumor clonal studies. Furthermore, our study warns about possible bias in commonly used experimental models where supposed highly malignant cell lines poorly represent the real cells able to drive tumorigenesis. Even if the first passage in immunodeficient mice is sufficient to induce important clonal selection, we provide evidence that OS can follow a neutral growth model where sub-clones can coexist and proliferate collectively at secondary transplantation (Figure 5).

## Figures and Tables

**Figure 1 cancers-13-02003-f001:**
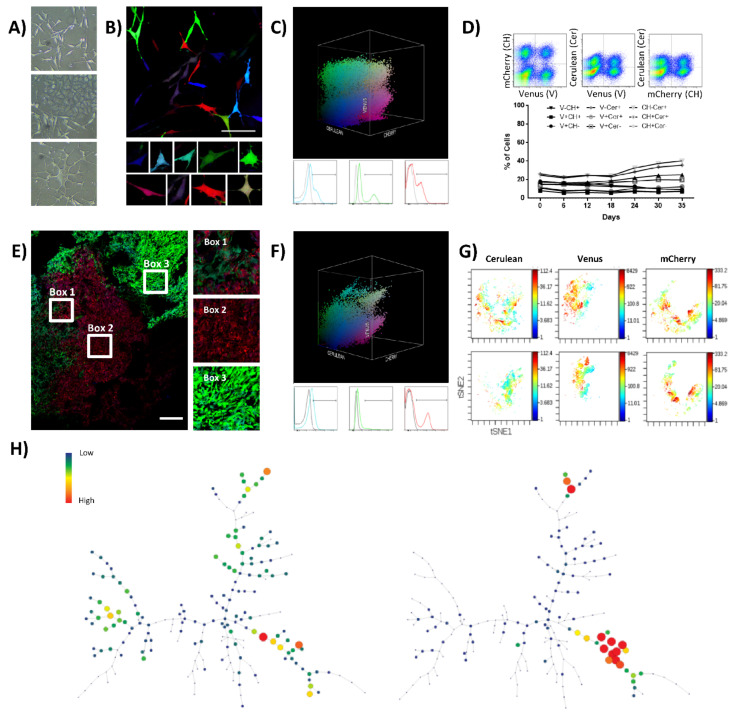
K5 OSA RGB cells develop oligoclonal tumors in vivo. (**A**) Phase-contrast microscopy images of K5 OSA cells; from the top to the bottom, representative images of cells with fibroblastic, tile-osteoblastic and multinucleated cell morphology during in vitro culturing. (**B**) Representative confocal microscopy images of K5-RGB cells in vitro; in the lower part, a detailed picture series of marked cells. Note that RGB-marking induces a highly variable colorimetric fluorescent marking; white bar = 100 µm. (**C**) Three-dimensional flow cytometry plot of K5-RGB cells in vitro; the three dimensions of the 3D dot plot represent the three RGB fluorescent variants: Cerulean, Venus and mCherry. Dots are colored according to their fluorescent marker expression or their possible combinations. In the lower part, histograms showing transduced cells vs. control (un-transduced K5 OSA cells); fluorescent marker signal distribution is presented separately in blue (Cerulean), green (Venus), and red (mCherry). For detailed MOI calculation and RGB-marking optimization, see Appendix A. (**D**) Fluorescent marker quantification of K5-RGB cells during 35 days of in vitro culture. In the upper part, the gating strategy is presented; the percentage of cells was quantified according to the 9 possible combinations of two fluorescent markers per time: Cerulean (Cer), Venus (V), and mCherry (CH). Statistical analysis indicated no correlation between population frequencies and day of culture. (**E**) Representative confocal microscopy image of K5-RGB tumors, note the clonal patches of colors. Detailed images are presented on the right; white bars = 250 µm. (**F**) Three-dimensional flow cytometry dot plot representation of explanted K5 RGB tumor cells. The three-dimensional dot plot is organized according to the three RGB fluorescent variants: Cerulean, Venus and mCherry. Dot color matches markers expression. In the lower part, histogram plots representing separately the fluorescent markers distribution. Note that clonal discrimination is difficult in histogram plots, while clonal populations are easily identified in the 3D representation. (**G**) Representative viSNE analysis of K5-RGB cells in vitro (upper part) and K5-RGB tumor cells ex vivo (lower part). The graphs show the enrichment and the loss of detection of some cell populations. In this case, the rainbow scale indicates cell frequency. (**H**) Representative SPADE analysis of K5-RGB cells in vitro (left) and K5-RGB tumor cells ex vivo (right); the picture shows the loss of detection of some populations. The complete 3D flow cytometry, viSNE and SPADE tree analyses are presented in Appendix A.

**Figure 2 cancers-13-02003-f002:**
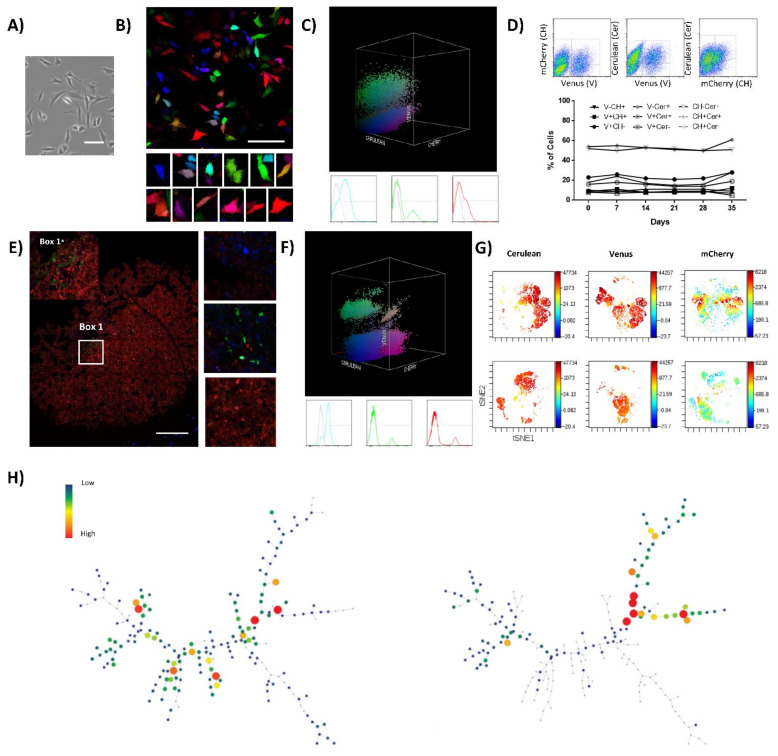
Saos2-RGB cells develop oligoclonal tumors in vivo. (**A**) Phase-contrast microscopy images of Saos2 cells; notice the tile-osteoblastic morphology during in vitro culture. White bars = 100 µm. (**B**) Representative confocal microscopy images of Saos2-RGB cells in vitro; in the lower part, a detailed picture series of marked cells. White bars = 100 µm. (**C**) Three-dimensional flow cytometry plot of Saos2-RGB cells in vitro; the three dimensions of the 3D dot plot represent the three RGB fluorescent variants: Cerulean, Venus and mCherry. Dots are colored according to their fluorescent marker expression or their possible combinations. In the lower part, histograms showing transduced cells vs. control (un-transduced Saos2 cells); fluorescent marker signal distribution is presented separately in blue (Cerulean), green (Venus), and red (mCherry). For detailed MOI calculation and RGB-marking optimization, see Appendix A. (**D**) Fluorescent marker quantification of Saos2-RGB cells during 35 days of in vitro culture. In the upper part, the flow cytometry gating strategy is presented; cells were quantified according to the possible combinations of two fluorescent markers per time: Cerulean (Cer), Venus (V), and mCherry (CH). Statistical analysis suggested no correlation between population frequencies and day of culture. (**E**) Representative confocal microscopy image of Saos2-RGB tumors, note that sub-clonal mixing is rare. Detailed images are presented on the right; white bar = 200 µm. (**F**) Three-dimensional flow cytometry dot plot representation of explanted Saos2-RGB tumor cells. Three-dimensional dot plot is organized according to the three RGB fluorescent variants: Cerulean, Venus, and mCherry. Dot colors match markers expression. In the lower part, histogram plots separately represent the fluorescent marker distribution. (**G**) Representative viSNE analysis of Saos2-RGB cells in vitro (upper part) and Saos2-RGB tumor cells ex vivo (lower part). The graphs show the enrichment and the loss of detection of some cell populations. In this case, the rainbow scale indicates cell frequency. (**H**) Representative SPADE analysis of Saos2-RGB cells in vitro (left) and Saos2-RGB tumor cells ex vivo (right); the picture shows the loss of detection of some populations. The complete 3D flow cytometry, viSNE, and SPADE tree analyses are presented in Appendix A.

**Figure 3 cancers-13-02003-f003:**
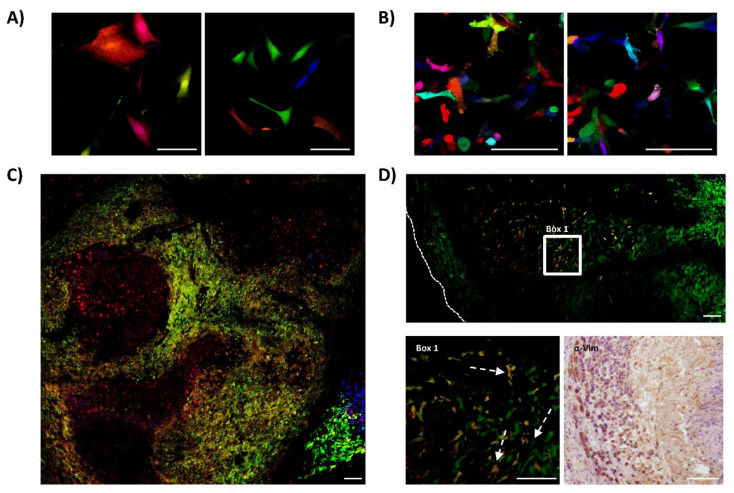
Canine and human primary OS cells contain few cells with tumor-propagating potential. The figure shows the results obtained from studying the clonal composition of tumors generated by canine (Laikos) and human (531B) primary samples in immunodeficient mice. (**A**) Representative confocal microscopy images of Laikos-RGB cells in vitro; white bars = 100 µm. (**B**) Representative confocal microscopy images of 531B-RGB cells in vitro; white bars = 100 µm. (**C**) Representative confocal microscopy image of Laikos-RGB tumors, note the globular clonal patches of colors; white bars = 100 µm. (**D**) Representative microscopy images of 531B-RGB tumors. At the top of the panel, a representative confocal microscopy tumor map. At the lower-left, a detailed view of the previous image (Box 1) showing the clonal growth of a bright green clone and some low-frequency clones of different clonal origin (white arrows). Human cell identity was confirmed by immunohistochemistry (anti-vimentin) and is shown at the lower-right. White bars = 100 µm.

**Figure 4 cancers-13-02003-f004:**
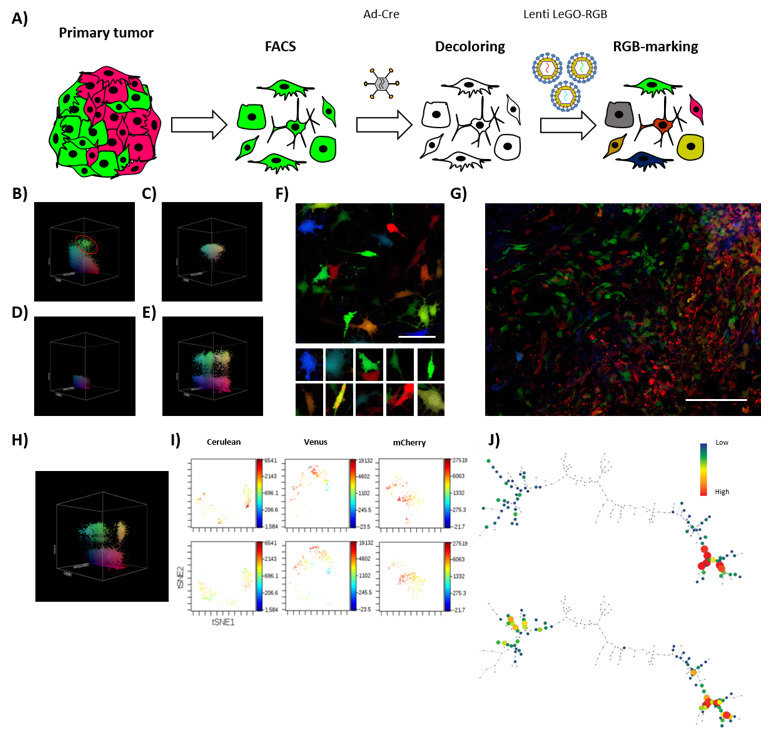
Tumor-derived clones grow neutrally at sub-clonal level. (**A**) The schematic shows the process of isolation, decoloring, and re-coloring, performed to study the sub-clonal dynamics of a clonal population isolated from primary tumors. (**B**) Representative 3D flow cytometry dot plot showing the clonal composition of K5 RGB-marked tumors at 35 days of primary transplantation; red circle indicates a FACS-grouped clone. The three dimensions of the 3D dot plot represent the three RGB fluorescent variants: Cerulean, Venus and mCherry. Dots are colored according to their fluorescent marker expression or their possible combinations. (**C**) Representative 3D flow cytometry dot plot showing a FACS-grouped tumor-derived clones after in vitro culture. (**D**) Representative 3D flow cytometry dot plot showing fluorescent marker expression loss after the Ad-Cre transduction of sorted tumor-derived clones. (**E**) Representative 3D flow cytometry dot plot showing fluorescent markers expression after RGB re-coloring of sorted tumor-derived clones. (**F**) Representative confocal microscopy image of re-traced tumor-derived clones; note the color and morphological heterogeneity in the picture series presented at the bottom of the figure. White bar: 50 µm. (**G**) Representative confocal microscopy image of tumors generated at secondary transplantation by retraced tumor-derived clones; note that both sub-clonal cell mixing (left) and an almost-homogeneous clonal development (right) occur. White bar: 100 µm. (**H**) Representative 3D flow cytometry dot plot showing fluorescent marker expression of tumor cells after secondary transplantation. (**I**) Representative viSNE analysis of retraced monoclonal population in vitro (day 0, top) and in vivo (day 20, bottom). (**J**) Representative SPADE analysis of K5 RGB cells in vitro (day 0, top) and in vivo (day 20, bottom). The complete viSNE and SPADE tree analyses are presented in Appendix A.

**Figure 5 cancers-13-02003-f005:**
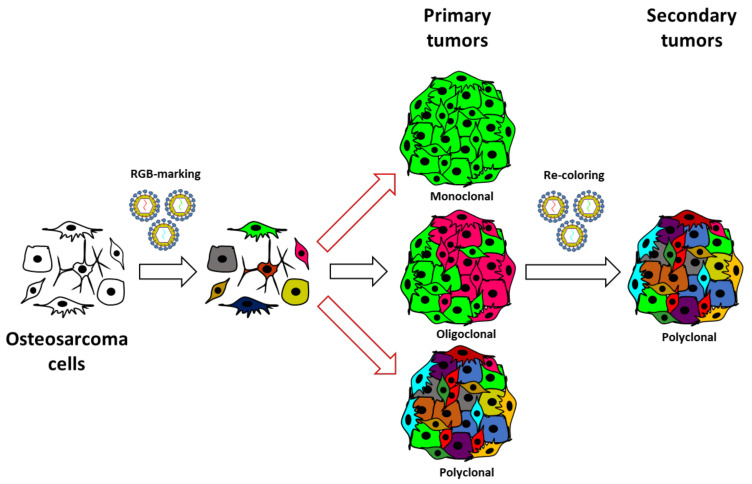
Revised model of OS growth. The schematic illustrates the experimental strategy and the clonal composition of induced OS through serial in vivo transplantation. Our study was based on the in vitro RGB marking of human, murine, and canine OS cells and in vivo evolution after their inoculation in immunodeficient mice. Our data indicate that selective events occur at primary transplantation, but when clonal populations overcome procedural selective forces, they follow a neutral growth model. This model agrees with the coexistence of different tumoral clones at primary transplantation and a sub-clonal heterogeneity at secondary transplantation.

## Data Availability

Given the nature of the study and methodology employed, the data supporting our findings are available from the corresponding author upon reasonable request.

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
