# Peer review of "RGB-Marking to Identify Patterns of Selection and Neutral Evolution in Human Osteosarcoma Models"

_cancers, 2021, doi:10.3390/cancers13092003_

Round 1

Reviewer 1 Report

In this manuscript the authors employed lentiviral-LeGO vectors to RGB-mark OS cells and studied tumor heterogeneity and in vivo clonal dynamics. They applied this method to study murine, canine and human OS samples, including cell lines and primary cultures, and focused on the tumor-propagating potential of the tumor cell lines and primary sample in vivo. By flow cytometry, confocal microscopy and different types of supervised and unsupervised clonal analyses, they found that TPP is low among cultured tumor cells, with evidence of strong selective clonal events occurring at tumor transplantation. They also demonstrate that OS can follow a neutral model of growth, where the disease is defined by the coexistence of different tumor subclones.   They concluded on the importance of rigorous testing selective forces in commonly used experimental models.

The method is of interest and the manuscript is well conducted and illustrated.

The authors could give more details on the NOD.Cg-Prkdcscid-Il2rgtm1Wjl/SzJ mice

Even if animal models represent a largely exploited tool for studying human cancers., they have got a lot of limits. This study demonstrates even more the heterogeneity of the in vivo clonal dynamic between xenografted cell lines (canine versus human or murine). They show that transposing the results obtained in vivo to what happens in patients is very random and requires perfect mastery of the conditions for producing xenografts, so as not to give erroneous interpretations of the results. For these reasons, it is important to end the manuscript with reference to recent articles carried out in the patient and which demonstrate the intratumoral heterogeneity of osteosarcomas. These recent studies reinforce the fact that animal models are only tools for studying tumors, especially since they are complex tumors such as osteosarcoma.

At the end of the manuscript, authors must mention 2 additional references, showing the role of the microenvironment in OS patient and discuss them

1) Zhou y et al: Single-cell RNA landscape of intratumoral heterogeneity and immunosuppressive microenvironment in advanced osteosarcoma nat comm 2020 Dec 10;11(1):6322. 

2) Gomez-Brouchet A et al : cancers 2021 ; Characterization of Macrophages and Osteoclasts in the Osteosarcoma Tumor Microenvironment at Diagnosis: New Perspective for Osteosarcoma Treatment?

After these corrections and clarifications, the manuscript can be published.

Author Response

Dear Editor,

Thank you very much for the manuscript review; Reviewers´ presented really interesting questions and comments which further open intriguing scientific discussions in our work. We are very pleased to see that both Reviewers considered our approach relevant, even if our work presents weaknesses intrinsic to the model. We greatly appreciate them for pointing out some shortcomings whose argumentations enriched the scientific level of our work.   

Attached you can find a Word with a point-by-point response to both reviewers.

Reviewer 2 Report

This paper describes a technique used to test clonal evolution of osteosarcoma over time. The authors use 3 cell lines, and tag cells with different colours, which they are able to track using FACS analysis. 

There are aspects to address.

This appears to be a methodology paper, however to someone who it is still unclear to me whether cells with specific markers were tagged and deemed to be separate clones. It is unclear whether cells were FACS sorted to have specific CD markers in specific clones. these were then tagged with different colours. This problem keeps arising, for instance in line 110 authors state ' discrete cell populations in vivo were also cell sorted...' what were they sorted into? Was it linked to the original population of cells which were colour-coded?

Authors used 3 different cell lines and reported specific clonal populations- however it is unclear what key features were specific and common between the clones?

The cells were implanted subcutaneously into murine models, however I believe an orthotopic implantation would have been more relevant. The native micro-environment will heavily influence the clonal evolution of cells and therefore when authors state that there was a consistent reduction in the clonal composition of RGB marked tumours, I feel the relevance is not explicit. 

Following on from this the tumour propagating potential was deemed scarce in the cell lines and the primary cells- could this just be the direct effect of a non-biomimetic environment?

I do think the technology is exciting and relevant- but I think the authors need to extensively re-write the paper to be more understandable and state limitations. 

Author Response

Dear Editor,

Thank you very much for the manuscript review; Reviewers´ presented really interesting questions and comments which further open intriguing scientific discussions in our work. We are very pleased to see that both Reviewers considered our approach relevant, even if our work presents weaknesses intrinsic to the model. We greatly appreciate them for pointing out some shortcomings whose argumentations enriched the scientific level of our work.    

Attached you can find a Word ith a point-by-point response to reviewers.

Round 2

Reviewer 2 Report

My comments have been addressed.